# Plasma Phospholipid Fatty Acids, *FADS1* and Risk of 15 Cardiovascular Diseases: A Mendelian Randomisation Study

**DOI:** 10.3390/nu11123001

**Published:** 2019-12-07

**Authors:** Shuai Yuan, Magnus Bäck, Maria Bruzelius, Amy M. Mason, Stephen Burgess, Susanna Larsson

**Affiliations:** 1Unit of Cardiovascular and Nutritional Epidemiology, Institute of Environmental Medicine, Karolinska Institutet, SE-171 77 Stockholm, Sweden; shuai.yuan@stud.ki.se; 2Department of Surgical Sciences, Uppsala University, SE-751 85 Uppsala, Sweden; 3Department of Medicine, Center for Molecular Medicine, Karolinska Institutet, SE-171 76 Stockholm, Sweden; magnus.back@ki.se; 4Heart and Vascular Theme—Division of Valvular and Coronary Disease, Karolinska University Hospital, SE-171 76 Stockholm, Sweden; 5Coagulation Unit, Department of Hematology, Karolinska University Hospital, SE-171 76 Stockholm, Sweden; maria.Bruzelius@ki.se; 6Department of Medicine Solna, Karolinska Institutet, SE-171 76 Stockholm, Sweden; 7Department of Public Health and Primary Care, University of Cambridge, CB1 8RN Cambridge, UK; am2609@medschl.cam.ac.uk (A.M.M.); sb452@medschl.cam.ac.uk (S.B.); 8MRC Biostatistics Unit, University of Cambridge, CB2 0SR Cambridge, UK

**Keywords:** cardiovascular disease, 𝛥5-desaturase, diet, *FADS*, fatty acids, Mendelian randomisation

## Abstract

Whether circulating fatty acids (FAs) play a causal role in the development of cardiovascular disease (CVD) remains unclear. We conducted a Mendelian randomisation study to explore the associations between plasma phospholipid FA levels and 15 CVDs. Summary-level data from the CARDIoGRAMplusC4D, MEGASTROKE, and Atrial Fibrillation consortia and UK Biobank were used. Sixteen single-nucleotide polymorphisms (SNPs) associated with ten plasma FAs were used as instrumental variables. SNPs in or close to the *FADS1* gene were associated with most FAs. We performed a secondary analysis of the association between a functional variant (rs174547) in *FADS1*, which encodes 𝛥5-desaturase (a key enzyme in the endogenous FA synthesis), and CVD. Genetic predisposition to higher plasma α-linolenic, linoleic, and oleic acid levels was associated with lower odds of large-artery stroke and venous thromboembolism, whereas higher arachidonic and stearic acid levels were associated with higher odds of these two CVDs. The associations were driven by SNPs in or close to *FADS1*. In the secondary analysis, the minor allele of rs174547 in *FADS1* was associated with significantly lower odds of any ischemic stroke, large-artery stroke, and venous thromboembolism and showed suggestive evidence of inverse association with coronary artery disease, abdominal aortic aneurysm and aortic valve stenosis. Genetically higher plasma α-linolenic, linoleic, and oleic acid levels are inversely associated with large-artery stroke and venous thromboembolism, whereas arachidonic and stearic acid levels are positively associated with these CVDs. The associations were driven by *FADS1*, which was also associated with other CVDs.

## 1. Introduction 

Cardiovascular disease (CVD) is one of the leading causes of mortality and disability among men and women worldwide [1,2]. It was estimated that CVD caused 16.7 deaths worldwide in 2010, and the number was projected to increase to 23.3 million in 2030 [1]. As a modifiable risk factor, suboptimal diet is an important preventable risk factor for CVD [3]. Among dietary factors, the types of fatty acids (FAs) consumed have been considered to play an important role in the prevention of CVD [4]. It has been proposed that increased intake of unsaturated FAs, particularly polyunsaturated fatty acids (PUFA), and decreased intake of saturated fatty acids (SFA) will decrease the incidence of CVD by lowering low-density lipoprotein cholesterol and triglyceride levels [4,5,6], improving insulin sensitivity [6], and reducing systemic inflammation [6]. However, available evidence on the associations of specific FAs with CVD risk is inconsistent or limited [7,8,9,10]. 

Fatty acid desaturase 1 (*FADS1*) gene encodes 𝛥5-desaturase, which is the key rate-limiting enzyme in the endogenous synthesis of long-chain PUFAs (Figure 1) [11]. 𝛥5-desaturase produces arachidonic acid (AA) and eicosapentaenoic acid (EPA) from dihomo-γ-linolenic acid and eicosatetraenoic acid, respectively [11]. Several observational studies have shown that 𝛥5-desaturase activity (measured by the ratio of AA to dihomo-γ-linolenic acid) is associated with the risk of metabolic syndrome [12], type 2 diabetes mellitus [13], and certain CVDs [14,15]. However, data on the associations of 𝛥5-desaturase activity or *FADS1* variants with CVDs are limited.

Mendelian randomisation (MR) is a method that strengthens causal inference between risk factors and outcomes by exploiting genetic variants as instrumental variables of an exposure [16]. This technique minimizes confounding as genetic variants are randomly assorted at meiosis, thereby having no association with self-selected lifestyle factors and behaviors. In addition, it overcomes reverse causality since allelic randomisation antedates the disease’s onset. 

We conducted a two-sample MR study to explore the associations of plasma phospholipid FA levels with risk of 15 CVDs. Since 𝛥5-desaturase plays a major role in FA synthesis, we performed a secondary analysis to investigate the associations of a functional variant (rs174547) within the *FADS1* gene, as a proxy for 𝛥5-desaturase activity, with CVD.

## 2. Methods

### 2.1. Study Design

We employed a two-sample MR design in the present study. Information of data sources used in this MR study and definitions for each CVD are summarized in Appendix A. Studies included in the genome-wide association studies (GWASs) had been approved by a relevant institutional review board and participants had provided informed consent. The present MR study has been approved by the Swedish Ethical Review Authority.

### 2.2. Selection of Plasma Fatty Acids and Instrumental Variables 

Selection of individual FAs was based on available data from meta-analyses of GWASs of 8631 to 8961 individuals of European ancestry (Appendix A) [17,18,19]. Ten individual FAs potentially associated with CVD were included in this MR study: α-linolenic acid (ALA); EPA; docosapentaenoic acid (DPA); docosahexaenoic acid (DHA); linoleic acid (LA); AA; palmitoleic acid (POA); oleic acid (OA); palmitic acid (PA) and stearic acid (SA). Since the concentrations (represented as the percentage of total FAs) of individual FAs vary largely, change in standard deviation (SD) was adapted as the unit. SDs for each FA were obtained from a population-based cohort study of middle-aged and older white participants of the Atherosclerosis Risk in Communities Study [20] and are provided in Appendix A. 

Information of the 16 single-nucleotide polymorphisms (SNPs) selected as instrumental variables for individual FAs in this MR study is summarized in Appendix A. All SNPs were associated with plasma FAs at the level of genome-wide significance (*p* < 5×10^−8^). Two SNPs (rs780093 and rs780094) located in the *GCKR* locus were excluded because *GCKR* is a highly pleiotropic locus associated with a number of phenotypes, such as body mass index, alcohol intake, and serum calcium levels, which are potential confounders in analyses of plasma FA levels and CVD. All SNPs for each individual FA were in different gene regions and in linkage equilibrium.

Rs174547 in the *FADS1* gene was used as a proxy for 𝛥5-desaturase activity in the secondary analysis. The minor (C) allele of rs174547 is associated with lower *FADS1* gene expression in human liver [21] and is also associated with plasma phospholipid PUFA levels in the direction anticipated from reduced 𝛥5-desaturase activity [22]. Rs174547 explained a large variation in levels of several FAs, especially AA (up to 37.6%) (Appendix A).

### 2.3. Outcome Data Sources

Summary-level data for the CVD outcomes were acquired from the CARDIoGRAMplusC4D consortium for coronary artery disease [23], the MEGASTROKE consortium for ischemic stroke and its subtypes [24], the Atrial Fibrillation consortium for AF [25], and the UK Biobank for nine other major CVDs (heart failure, aortic valve stenosis, abdominal and thoracic aortic aneurysms, transient ischemic attack, intracerebral and subarachnoid hemorrhages, venous thromboembolism, and peripheral vascular disease) [26] (Appendix A). Our analyses of data from the UK Biobank included 367,643 participants after exclusion of non-European individuals (to reduce population stratification), related individuals (third-degree relatives or closer), low call rate, and excess heterozygosity (3 or more standard deviations from the mean).

### 2.4. Statistical Analysis

The conventional inverse-variance weighted method with fixed-effects was used to analyze the associations between plasma FA levels and 15 CVDs. The results reported represent the odds ratios (ORs) for an SD increase in genetically predicted plasma FA levels. In sensitivity analyses, we employed SNPs, except variants in or close to the *FADS1* gene, as instrumental variables for FAs to exclude the driving effect of FADS1. In the secondary analysis of 𝛥5-desaturase activity, ORs with 95% confidence intervals (CI) for each CVD per additional C allele of rs174547 in *FADS1* were derived from the log ORs (beta-coefficients for the SNP-CVD associations) and standard errors. To account for multiple testing, we considered associations with *p* values below 3.33×10^−4^ (where *p* = 0.05/150 (10 FAs and 15 CVDs)) to represent strong evidence of causal associations in the main analysis, and *p* values below 0.003 (*p* = 0.05/15 CVDs) to represent statistical significance in the secondary analysis of rs174547. Associations with *p* values below 0.05 but above 3.33×10^−4^ and 0.003 in the main and secondary analyses, respectively, were regarded as suggestive evidence of associations. All statistical analyses were performed in Stata/SE 15.0.

### 2.5. Pleiotropy Assessment

We used the PhenoScanner V2 database [27] to explore pleiotropic associations of the FA-associated SNPs with other traits at the genome-wide significance level. Similar or related traits were reported and documented.

## 3. Results

### 3.1. Plasma FAs and CVDs

The associations between genetically predicted plasma FAs and the 15 CVDs are displayed in Table 1 and Figure 2. Genetic predisposition to higher plasma levels of ALA, LA, and OA was significantly or borderline significantly (*p* = 5.34×10^−4^) associated with lower odds of large-artery stroke and venous thromboembolism, whereas higher plasma AA and SA (only for venous thromboembolism) levels were significantly or borderline significantly (*p* = 3.34×10^−4^) associated with higher odds of these two CVDs. There was suggestive evidence of inverse associations of genetically predicted ALA, LA, OA, or POA levels with one or more of the other 10 CVDs, as well as positive associations of genetically predicted EPA, DPA, DHA, AA, or SA levels with one or more CVD. However, only a significant association between DHA and subarachnoid hemorrhage was still obtained when SNPs in or close to the *FADS1* gene (rs174547, rs174538, and rs102275) were not included in the analysis.

### 3.2. FADS1 and CVD

There was a statistically significant inverse association between the minor allele of rs174547 (minor allele frequency of 0.34 in UK Biobank) in *FADS1* and three of the 15 CVDs, including any ischemic stroke, large-artery stroke, and venous thromboembolism (Figure 3). Moreover, there was a suggestive inverse association of rs174547 with coronary artery disease, abdominal aortic aneurysm and aortic valve stenosis (Figure 3).

### 3.3. Pleotropic Associations

Pleiotropic associations of FA-associated SNPs with other traits are shown in Appendix A. SNPs in or close to the *FADS1* gene were significantly associated with several traits, including triglyceride, cholesterol and fasting glucose levels, blood cells, height, pulse rate, and heart rate. Rs16966952 in *PDXDC1* was associated with fat-free mass, height, and weight. Rs10740118 in *JMJD1C* was associated with triglyceride levels, blood cells, diastolic blood pressure, body mass index, and education. The SNPs rs603424 nearby *SCD* and rs11190604 in *HIF1AN* were associated with blood cells and fat-free mass, respectively.

## 4. Discussion

The main finding of this MR study is that genetically higher plasma levels of ALA, LA, and OA were associated with lower odds of large-artery stroke and venous thromboembolism, whereas higher plasma AA and SA levels were associated with higher odds of these two CVDs. However, these and other suggestive associations of plasma levels of several FAs with CVD were driven by *FADS1*, encoding the 𝛥5-desaturase enzyme, which showed strong or suggestive associations with six out of 15 CVDs. 

### 4.1. Plasma FAs and Ischemic Stroke

Observational studies of the association between individual FAs and ischemic stroke, including large-artery stroke, are relatively limited and the findings are conflicting, possibly because of small sample sizes of some studies and residual confounding. Most but not all studies support that moderate or high ALA [28] and LA [29,30] and low SA [31,32] exposure is associated with lower risk of ischemic stroke, which is in line with our results, although our findings for SA were not significant. A large Danish cohort study reported a significant inverse association between the content of LA in adipose tissue and risk of large-artery stroke [30]. However, circulating ALA levels were not significantly associated with ischemic stroke in smaller cohorts of Swedish [33] and U.S. [34] adults. With regard to MUFAs, the findings vary considerably between available studies. In two cohort studies conducted in U.S. populations, serum POA and OA levels were positively associated with risk of ischemic stroke [34,35]. However, a null association of those FAs with ischemic stroke was reported in two systematic reviews [36,37]. In our study, we found a suggestive inverse association of higher plasma OA levels with any ischemic stroke and large-artery stroke in particular. Support for a possible role of MUFAs and PUFAs for prevention of stroke comes from a large-scale randomized trial in Spain, which showed that a Mediterranean diet supplemented with either olive oil (high in OA and LA) or nuts (high in MUFAs and PUFAs) significantly reduced the incidence of total stroke [38]. However, another finding from that trial revealed that extra virgin olive oil but not normal olive oil exerted a protective effect on cardiovascular disease [39], suggesting that other minor components of extra virgin olive oil, rather than OA, explained the protective effect.

### 4.2. Plasma FAs and Venous Thromboembolism

Data on the associations between plasma levels of FAs and venous thromboembolism are scarce and inconsistent. In two reviews, although higher n-3 PUFAs appear to influence collagen-induced platelet aggregation in a favorable manner, there is no clear evidence of beneficial effects on fibrinolysis and blood coagulability [40,41]. Another systematic review on coronary thrombosis pointed out that the beneficial effect of n-3 PUFAs on coronary heart disease may be antiarrhythmic rather than antithrombotic [42]. In animal studies, EPA showed a potential effect against thrombosis in rats [43], and results of a cohort study of Norwegian adults showed that dietary intake of marine n-3 PUFAs was inversely associated with risk of venous thromboembolism [10] and risk of recurrent venous thromboembolism after unprovoked index events [44]. However, if anything, we observed a detrimental effect of genetically higher EPA levels on venous thromboembolism in the present study. More studies are needed to determine the association between FAs and venous thromboembolism. In addition, we observed a positive association between SA and venous thromboembolism. In vivo, SA has been found to be easily converted into OA [45], which showed an inverse association with venous thromboembolism in the present study. Thus, whether the observed positive association between SA and venous thromboembolism is direct or mediated by OA remains unclear. Previous studies on SA showed a neutral or unclear effect on plasma lipids [46]. Elucidation of the detailed mechanism behind the association between SA and venous thromboembolism warrants further study. 

### 4.3. Plasma FAs and other CVDs

The present study found several suggestive associations between genetically predicted levels of certain FAs and CVDs, such as inverse associations of higher levels of ALA, LA, OA, and POA as well as lower levels of EPA, DPA, DHA, AA, or SA levels with coronary artery disease, aortic valve stenosis and abdominal aneurysm. The observed effects of different FAs on coronary artery disease were, overall, in line with observational studies [47,48]. Nonetheless, epidemiological studies focusing on serum FAs and aortic valve stenosis and abdominal aneurysm are limited. 

With regard to atrial fibrillation, observational studies consistently concluded a null association between n-3 PUFAs and incidence of atrial fibrillation [49], which is supported by our findings. In addition, among patients undergoing cardiac surgery, supplementation of n-3 PUFAs did not decrease the risk of post-operative atrial fibrillation [50]. 

### 4.4. FADS1 and CVD

To the best of our knowledge, this is the first study to examine and show that a functional variant in *FADS1* is inversely associated with abdominal aortic aneurysm, aortic valve stenosis, large-artery stroke and venous thromboembolism. In an experimental study, we found that the minor allele of rs174547 is associated with increased desaturase activity in the n-3 PUFA pathway in human aortic valves, leading to increased DHA content [51]. In the present MR study, we could not assess the role of tissue levels of DHA, but plasma DHA levels were inversely associated with subarachnoid hemorrhage only.

Our results for *FADS1* based on data from the CARDIoGRAMplusC4D consortium are in line with those of some prior smaller studies of the association of genetic variations in *FADS1* or 𝛥5-desaturase activity with coronary artery disease. A case-control study, comprising 1646 adults (756 coronary artery disease cases), found that the T allele of rs174556 (strongly correlated with the C allele of rs174547) was associated with a lower risk of coronary artery disease [52]. This finding was verified by another study in an Asian population, including 505 cases and 510 controls, which found that low 𝛥5-desaturase activity was associated with lower odds of coronary artery disease [53]. In contrast, in a case-control study of 2448 postmenopausal women, 𝛥5-desaturase activity was inversely associated with coronary artery disease [54], and a small case-control study found that the frequency of the T allele of rs174556 (correlated with the C allele of rs174547) was higher among the cases than among the controls [55]. Other studies, including two prospective studies [56,57] (of which one relatively large cohort of 24,032 Swedish adults [57]) and a case-control study [58] found no significant association of SNPs in *FADS1* or 𝛥5-desaturase activity (defined by the ratio of AA to dihomo-γ-linolenic acid) with risk of coronary artery disease or ischemic stroke. The inconsistency may be explained by the small sample sizes in most previous studies, by differences in proportion of n-3 and n-6 PUFAs in the diet in different populations, or by synchronization of low 𝛥5-desaturase activity and low 𝛥6-desaturase activity.

### 4.5. Possible Mechanisms

Variants in or near the *FADS1* gene are associated with several potential intermediates, including cholesterol and triglyceride levels and heart rate. Meta-analyses of randomized controlled trials have shown that n-3 PUFA supplementation has a beneficial effect on cholesterol and triglyceride levels as well as heart rate variability [48,59]. In addition, evidence from short-term feeding trials indicates that replacement of SFAs with PUFAs or MUFAs lowers low-density lipoprotein cholesterol and triglyceride levels [5,6]. Animal and cell studies have demonstrated that FAs exert effects on lipid levels by influencing low-density lipoprotein receptor and turnover, very low-density lipoprotein secretion, lipogenesis, cholesterol 7α-hydroxylase activity, lipoprotein lipase activity and reverse cholesterol transport [60]. Therefore, plasma FAs levels play an important role in the development of CVDs via the regulation of these intermediates, in particular lipid levels.

There may be additional mechanisms whereby FAs may affect the risk of CVD, such as effects on inflammation [61,62,63] and oxidative stress [64]. 𝛥5-desaturase produces AA and EPA from dihomo-𝛾-linolenic acid and eicosatetraenoic acid, respectively [10]. AA-derived eicosanoids initiate and augment pro-inflammatory responses, whereas EPA-derived eicosanoids are less inflammatory or anti-inflammatory [65,66] and participate in the resolution of inflammation. FA-related eicosanoids may increase atherosclerotic plaque deposition, thereby facilitating atherosclerosis formation. Atherosclerosis and thrombosis development are vital pathological processes in the development of coronary artery disease and ischemic stroke [63]. In addition, in the metabolism of certain FAs (such as AA) through the cyclooxygenase pathways, excessive reactive oxygen species are generated [64]. These free radicals may damage vascular function, increase endothelial permeability, alter reactivity to vasodilators, and promote formation of focal lesions in endothelial cell membranes at very low levels by increasing vasodilation and platelet aggregability [63,64]. 

### 4.6. Strengths and Limiations

A major strength of this study is the MR approach, which minimizes confounding and reverse causality, potentially distorting the results of observational studies. In addition, we tested the association between plasma FA levels and CVD using summary-level data from large-scale genetic consortia or studies with large sample size, thereby guaranteeing high statistical power to detect weak associations. However, for certain CVDs, we had a limited number of cases, which might preclude inference about the absence of association in cases where no association was detected, such as for thoracic aortic aneurysm, intracerebral hemorrhage, and subarachnoid hemorrhage. The study populations mainly included individuals of European ancestry. Thus, bias due to population stratification was reduced. Nonetheless, population confinement limited the generalizability of our findings to other populations. Another limitation in the present study is that we only had one to four SNPs as instrumental variables for individual FAs. This confined sensitivity analyses to explore pleiotropy, which might potentially weaken the reliability of the results. However, we manually searched pleiotropic traits related to FAs. Some of the SNPs had pleiotropic associations with other traits but the SNPs may affect those traits through vertical (mediated) pleiotropy rather than through horizontal pleiotropy. A further major shortcoming is that SNPs in or close to the *FADS1* gene were associated with most FAs, which explained the major proportion of variance in most plasma FA levels, and were driving the associations. 

## 5. Conclusions

This study showed that genetic predisposition to higher plasma ALA, LA, and OA levels was associated with lower odds of large-vessel stroke and venous thromboembolism, whereas plasma AA and SA levels were positively associated with these CVDs. However, the associations were driven by *FADS1*, which was also significantly or suggestively associated with coronary artery disease, abdominal aortic aneurysm and aortic valve stenosis. Thus, it is recommended to reconsider the role of individual FAs in the prevention of different CVDs. 

## Figures and Tables

**Figure 1 nutrients-11-03001-f001:**
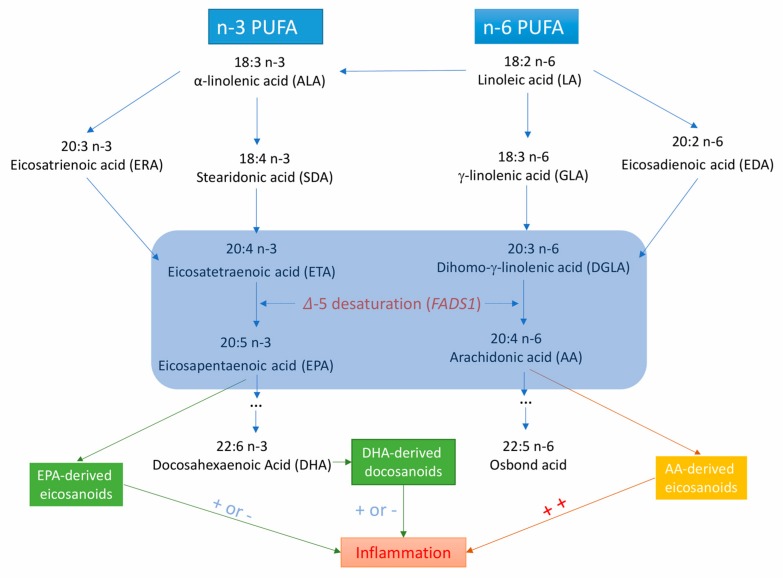
Metabolic and inflammation-related pathways for long-chain polyunsaturated fatty acids from the essential ɑ-linolenic acid and linoleic acid. ++ indicates that AA-derived eicosanoids are strong pro-inflammatory factors; + or − means EPA- and DHA-derived eicosanoids or docosanoids are less inflammatory or anti-inflammatory.

**Figure 2 nutrients-11-03001-f002:**
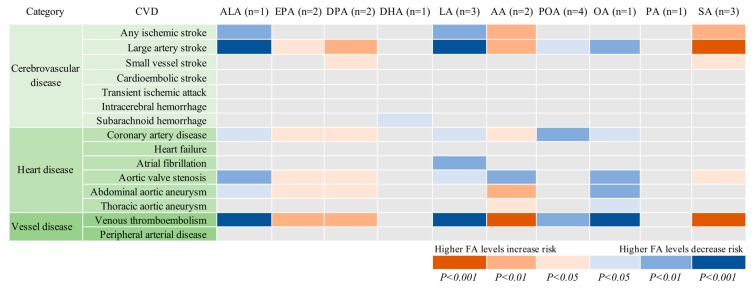
Associations of plasma phospholipid fatty acids levels with 15 cardiovascular diseases. AA indicates arachidonic acid; ALA, α-linolenic acid; DHA, docosahexaenoic acid; DPA, docosapentaenoic acid; EPA, eicosapentaenoic acid; LA, linoleic acid; OA, oleic acid; PA, palmitic acid; POA, palmitoleic acid; SA, stearic acid. The number in parenthesis after each fatty acid represents the number of single-nucleotide polymorphisms used as instrumental variables for each fatty acid.

**Figure 3 nutrients-11-03001-f003:**
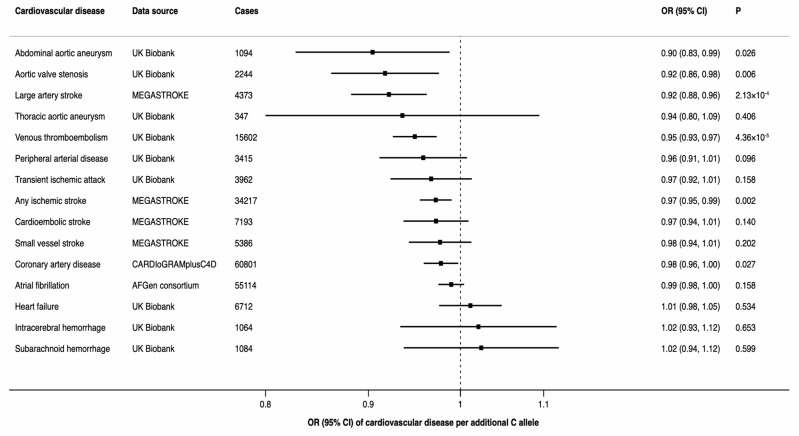
Associations between rs174547 in the *FADS1* gene and odds ratio of 15 cardiovascular diseases. CI indicates confidence interval; OR, odds ratio.

**Table 1 nutrients-11-03001-t001:** Associations of genetically predicted plasma fatty acid levels with 15 cardiovascular diseases.

**CVD**	**Cases**	**Dataset**	**ALA (*n* = 1 SNP)**		**EPA (*n* = 2 SNPs)**		**DPA (*n* = 2 SNPs)**		**DHA (*n* = 1 SNP)**	**LA (*n* = 3 SNPs)**		
			**OR (95% CI)**	***P* value**	**OR (95% CI)**	***P* value**	**OR (95% CI)**	***P* value**	**OR (95% CI)**	***P* value**	**OR (95% CI)**	***P* value**
**Cerebrovascular disease**												
Any ischemic stroke	60 341	MEGASTROKE	0.93 (0.89, 0.98)	0.002	1.06 (0.99, 1.13)	0.081	1.04 (1.00, 1.07)	0.060	1.11 (0.96, 1.27)	0.153	0.96 (0.93, 0.99)	0.006
Large artery stroke	6688	MEGASTROKE	0.82 (0.73, 0.91)	2.11×10^−4^	1.17 (1.01, 1.36)	0.035	1.13 (1.03, 1.23)	0.007	1.28 (0.92, 1.78)	0.142	0.86 (0.80, 0.93)	8.61×10^−5^
Small vessel stroke	11 710	MEGASTROKE	0.94 (0.86, 1.03)	0.203	1.13 (1.00, 1.27)	0.060	1.07 (1.00, 1.15)	0.084	0.85 (0.64, 1.12)	0.240	0.99 (0.93, 1.05)	0.746
Cardioembolic stroke	9006	MEGASTROKE	0.93 (0.85, 1.02)	0.141	1.04 (0.91, 1.18)	0.567	1.04 (0.97, 1.12)	0.284	1.05 (0.80, 1.40)	0.715	0.95 (0.89, 1.02)	0.149
Transient ischemic attack	3962	UK Biobank	0.92 (0.82, 1.03)	0.158	1.12 (0.95, 1.32)	0.174	1.05 (0.96, 1.15)	0.317	1.09 (0,76, 1.57)	0.631	0.96 (0.88, 1.04)	0.297
Intracerebral hemorrhage	1064	UK Biobank	1.05 (0.84, 1.32)	0.652	0.90 (0.66, 1.23)	0.506	1.00 (0.84, 1.19)	0.993	0.74 (0.37, 1.48)	0.387	1.03 (0.88, 1.20)	0.737
Subarachnoid hemorrhage	1084	UK Biobank	1.06 (0.85, 1.33)	0.599	1.07 (0.79, 1.46)	0.668	1.06 (0.89, 1.26)	0.549	0.47 (0.23, 0.93)	0.030 *	1.08 (0.93, 1.27)	0.317
**Heart disease**												
Coronary artery disease	60 801	CARDIoGRAMplusC4D	0.95 (0.90, 0.99)	0.027	1.08 (1.01, 1.15)	0.038	1.04 (1.00, 1.08)	0.045	1.00 (0.86, 1.16)	1.000	0.96 (0.92, 0.99)	0.011
Heart failure	6712	UK Biobank	1.03 (0.94, 1.13)	0.533	1.01 (0.89, 1.15)	0.866	0.99 (0.92, 1.06)	0.762	0.90 (0.68, 1.19)	0.463	1.02 (0.96, 1.08)	0.604
Atrial fibrillation	65 446	AFGen	0.97 (0.94, 1.01)	0.158	1.05 (0.99, 1.10)	0.085	1.02 (0.99, 1.05)	0.265	0.99 (0.88, 1.11)	0.813	0.97 (0.94, 0.99)	0.009
Aortic valve stenosis	2244	UK Biobank	0.81 (0.69, 0.94)	0.006	1.32 (1.07, 1.64)	0.011	1.17 (1.04, 1.33)	0.011	0.93 (0.58, 1.51)	0.779	0.88 (0.79, 0.98)	0.024
Abdominal aortic aneurysm	1094	UK Biobank	0.78 (0.62, 0.97)	0.025	1.41 (1.04, 1.92)	0.027	1.26 (1.06, 1.50)	0.010	0.60 (0.30, 1.19)	0.140	0.87 (0.74, 1.02)	0.077
Thoracic aortic aneurysm	347	UK Biobank	0.85 (0.57, 1.25)	0.406	1.04 (0.60, 1.79)	0.895	1.05 (0.77, 1.43)	0.761	1.52 (0.45, 5.14)	0.504	0.95 (0.72, 1.25)	0.708
**Vessel disease**												
Peripheral arterial disease	3415	UK Biobank	0.90 (0.79, 1.02)	0.096	1.18 (1.00, 1.41)	0.058	1.09 (0.99, 1.21)	0.085	0.93 (0.63, 1.37)	0.696	0.93 (0.85, 1.01)	0.087
Venous thromboembolism	15 602	UK Biobank	0.88 (0.82, 0.93)	4.39×10^−5^	1.13 (1.03, 1.23)	0.007	1.08 (1.03, 1.13)	0.003	1.14 (0.93, 1.38)	0.202	0.92 (0.88, 0.96)	1.95×10^−4^
**CVD**	**Cases**	**Dataset**	**AA (*n* = 2 SNPs)**		**POA (*n* = 4 SNPs)**		**OA (*n* = 1 SNP)**		**PA (*n* = 1 SNP)**		**SA (*n* = 3 SNPs)**	
			**OR (95% CI)**	***P* value**	**OR (95% CI)**	***P* value**	**OR (95% CI)**	***P* value**	**OR (95% CI)**	***P* value**	**OR (95% CI)**	***P* value**
**Cerebrovascular disease**												
Any ischemic stroke	60 341	MEGASTROKE	1.03 (1.01, 1.06)	0.002	0.97 (0.90, 1.05)	0.422	0.87 (0.80, 0.95)	0.002	0.90 (0.77, 1.05)	0.164	1.12 (1.04, 1.21)	0.003
Large artery stroke	6688	MEGASTROKE	1.10 (1.04, 1.15)	3.34×10^−4^	0.82 (0.68, 0.98)	0.031	0.68 (0.55, 0.85)	5.34×10^−4^	0.90 (0.62, 1.32)	0.591	1.35 (1.12, 1.61)	0.001
Small vessel stroke	11 710	MEGASTROKE	1.03 (0.99, 1.08)	0.136	0.95 (0.84, 1.09)	0.464	0.89 (0.75, 1.06)	0.196	0.86 (0.62, 1.18)	0.345	1.19 (1.03, 1.39)	0.021
Cardioembolic stroke	9006	MEGASTROKE	1.04 (0.99, 1.08)	0.099	0.90 (0.76, 1.06)	0.210	0.84 (0.69, 1.01)	0.057	0.98 (0.71, 1.34)	0.874	1.08 (0.92, 1.26)	0.344
Transient ischemic attack	3962	UK Biobank	1.04 (0.99, 1.10)	0.144	0.96 (0.77, 1.19)	0.687	0.84 (0.66, 1.06)	0.139	1.05 (0.70, 1.58)	0.805	1.11 (0.91, 1.36)	0.285
Intracerebral hemorrhage	1064	UK Biobank	0.97 (0.88, 1.08)	0.605	0.88 (0.59, 1.32)	0.526	1.15 (0.73, 1.82)	0.535	1.32 (0.61, 2.89)	0.482	0.81 (0.56, 1.19)	0.281
Subarachnoid hemorrhage	1084	UK Biobank	0.98 (0.88, 1.08)	0.673	1.18 (0.78, 1.77)	0.441	1.20 (0.76, 1.87)	0.437	0.78 (0.36, 1.69)	0.528	1.19 (0.82, 1.73)	0.372
**Heart disease**												
Coronary artery disease	60 801	CARDIoGRAMplusC4D	1.02 (1.00, 1.05)	0.042	0.89 (0.82, 0.97)	0.008	0.90 (0.81, 0.99)	0.036	1.03 (0.88, 1.22)	0.691	1.02 (0.94, 1.11)	0.620
Heart failure	6712	UK Biobank	0.99 (0.95, 1.03)	0.535	0.89 (0.76, 1.05)	0.180	1.06 (0.89, 1.27)	0.524	0.93 (0.68, 1.27)	0.631	1.02 (0.87, 1.18)	0.825
Atrial fibrillation	65 446	AFGen	1.01 (1.00, 1.03)	0.180	1.04 (0.97, 1.11)	0.262	0.95 (0.88, 1.02)	0.137	1.12 (0.98, 1.28)	0.087	1.01 (0.95, 1.08)	0.748
Aortic valve stenosis	2244	UK Biobank	1.11 (1.03, 1.19)	0.004	0.80 (0.60, 1.07)	0.128	0.66 (0.48, 0.90)	0.008	0.91 (0.53, 1.55)	0.724	1.33 (1.03, 1.73)	0.031
Abdominal aortic aneurysm	1094	UK Biobank	1.13 (1.02, 1.25)	0.022	0.88 (0.59, 1.32)	0.537	0.61 (0.39, 0.96)	0.032	1.39 (0.64, 2.99)	0.405	1.23 (0.85, 1.78)	0.282
Thoracic aortic aneurysm	347	UK Biobank	1.08 (0.90, 1.30)	0.388	1.07 (0.52, 2.21)	0.852	0.70 (0.32, 1.55)	0.376	1.03 (0.26, 4.04)	0.965	1.26 (0.65, 2.44)	0.501
**Vessel disease**												
Peripheral arterial disease	3415	UK Biobank	1.05 (0.99, 1.11)	0.104	0.92 (073, 1.16)	0.493	0.82 (0.64, 1.06)	0.135	1.06 (0.68, 1.64)	0.803	1.14 (0.92, 1.40)	0.242
Venous thromboembolism	15 602	UK Biobank	1.07 (1.04, 1.10)	1.90×10^−5^	0.85 (0.76, 0.95)	0.006	0.77 (0.68, 0.87)	4.09×10^−5^	0.92 (0.74, 1.14)	0.446	1.22 (1.10, 1.36)	2.32×10^−4^

AA indicates arachidonic acid; ALA, α-linolenic acid; CVD, cardiovascular disease; CI, confidence interval; DHA, docosahexaenoic acid; DPA, docosapentaenoic acid; EPA, eicosapentaenoic acid; LA, linoleic acid; OA, oleic acid; OR, odds ratio; PA, palmitic acid; POA, palmitoleic acid; SA, stearic acid; SNP, single-nucleotide polymorphism.* The significance remained in the sensitivity analysis excluding SNPs in or close to the *FADS1* gene.

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
