# Peer review of "Plasma Phospholipid Fatty Acids, FADS1 and Risk of 15 Cardiovascular Diseases: A Mendelian Randomisation Study"

_nutrients, 2019, doi:10.3390/nu11123001_

Round 1
Reviewer 1 Report
This well written paper discusses the association, evaluated through a MR approach, between plasma levels of specific fatty acids and CVD events risk.
I have only minor comments or issue to raise.
The association of oleic acid and CVD, as the authors state, is neutral in most studies. They cite the PREDIMED study as a support to a protective role of this FA (together with LA) against stroke. A secondary paper from this study (Guasch-Ferrè M et al, BMC Medicine 2014), on the other hand, shows that extravirgin olive oil exerts a protective effect, while normal olive oil does not. This suggests that minor components of EVOO, rahter than oleic acid, might explain the protection observed. Since oleic acid, additionally, has no clear effect on plasma lipids and lipoproteins, its possible protective role, observed in this study, should be discussed into a greater detail.
Stearic acid (positively associated with risk in this study) is easily converted, in vivo, into oleic acid (negatively associated with events in the present study), and has no clear effect on plasma lipids: this should also be discussed.
Minor comments: the first sentence of the introduction. could be improved; a fullstop is missing in the discussione after ref (5,6).
Author Response
Response to Reviewer 1
This well written paper discusses the association, evaluated through a MR approach, between plasma levels of specific fatty acids and CVD events risk.
I have only minor comments or issue to raise.
Point 1: The association of oleic acid and CVD, as the authors state, is neutral in most studies. They cite the PREDIMED study as a support to a protective role of this FA (together with LA) against stroke. A secondary paper from this study (Guasch-Ferrè M et al, BMC Medicine 2014), on the other hand, shows that extravirgin olive oil exerts a protective effect, while normal olive oil does not. This suggests that minor components of EVOO, rather than oleic acid, might explain the protection observed. Since oleic acid, additionally, has no clear effect on plasma lipids and lipoproteins, its possible protective role, observed in this study, should be discussed into a greater detail.
Response 1: Thank you for the comment. We agree that the established effects of oleic acid on stroke in the present study needs cautious interpretation given that no protective effect was observed for normal olive oil in the PREDIMED study. We have added the following text in the Discussion part (Page 8).
“However, another finding from that trial revealed that extra virgin olive oil but not normal olive oil exerted a protective effect on cardiovascular disease [39], suggesting that other minor components of extra virgin olive oil, rather than OA, explained the protective effect.”
Point 2: Stearic acid (positively associated with risk in this study) is easily converted, in vivo, into oleic acid (negatively associated with events in the present study) and has no clear effect on plasma lipids: this should also be discussed.
Response 2: Thank you for the comment. We have added several sentences to elaborate the association between stearic acid and oleic acid in vivo and discuss it when interpreting the association between stearic acid and venous thromboembolism. (Page 8)
“In addition, we observed a positive association between SA and venous thromboembolism. In vivo, SA has been found to be easily converted into OA [45], which showed an inverse association with venous thromboembolism in the present study. Thus, whether the observed positive association between SA and venous thromboembolism is direct or mediated by OA remains unclear. Previous studies on SA showed a neutral or unclear effect on plasma lipids [46]. Detailed mechanism behind the association between SA and venous thromboembolism warrants further study.”
Point 3: Minor comments: the first sentence of the introduction could be improved; a fullstop is missing in the discussion after ref (5,6).
Response 3: Thank you for comments. The first sentence of the introduction has been improved and a fullstop has been added correspondingly.
Introduction (Page 1): “Cardiovascular disease (CVD) is the leading cause of mortality and disability among men and women worldwide [1, 2]. It was estimated that CVD caused 16.7 deaths worldwide in 2010, and the number was projected to increase to 23.3 million in 2030 [1].”
Reviewer 2 Report
The authors of the paper presented a very interesting statistical study that allows to correlate Plasma Phospholipid Fatty Acids to 15 Cardiovascular Diseases. Moreover they show the first study able to examine and show that a functional variant in FADS1 is inversely associated with abdominal aortic aneurysm, aortic valve stenosis, large artery stroke and venous thromboembolism.
Author Response
Response to Reviewer 2
The authors of the paper presented a very interesting statistical study that allows to correlate Plasma Phospholipid Fatty Acids to 15 Cardiovascular Diseases. Moreover, they show the first study able to examine and show that a functional variant in FADS1 is inversely associated with abdominal aortic aneurysm, aortic valve stenosis, large artery stroke and venous thromboembolism.
Response 1: Thank you for the comment.
Reviewer 3 Report
This study adopted a strong design and examined the association between multiple fatty acids and a broad spectrum of cardiovascular diseases. The results are reasonable, and the conclusion is overall solid.
A major issue is that the selected 15 cardiovascular diseases are heterogeneous. It is actually a strength of this study to examine multiple disease outcomes, but a reasonable categorization and interpretation of their mechanisms are needed in such case for an article not to lose its focus. For example the pathophysiology of and risk factors for atrial fibrillation and venous thromboembolism (does this include deep/superficial venous thrombosis as well as pulmonary embolism?) are quite different from CAD/stroke/aneurysm/PAD. Hemorrhagic stroke and ischemic stroke are also different. Within the current results presentation format (e.g. Table 1), the 15 diseases are listed in a disorganized way, and the discussion are not streamlined to disease entities either (there is a focused paragraph for venous thromboembolism though). Also, transient ischemic attack should be grouped and discussed together with ischemic stroke, not separately.
Minor issues:
3rd paragraph (Line numbers after Line 148 are missing) in [Discussion]: you might want to include this updated study as well: Isaksen T, et al. 2019. Thromb Haemost (PMID: 31659738).
Discussion about atrial fibrillation is totally missing.
Line 9 of the last paragraph in Discussion: transferability or generalizability?
Last sentence of conclusion: it is farfetched to reach this conclusion as no therapy has been mentioned throughout the whole article.
Author Response
Response to Reviewer 3
This study adopted a strong design and examined the association between multiple fatty acids and a broad spectrum of cardiovascular diseases. The results are reasonable, and the conclusion is overall solid.
Point 1: A major issue is that the selected 15 cardiovascular diseases are heterogeneous. It is actually a strength of this study to examine multiple disease outcomes, but a reasonable categorization and interpretation of their mechanisms are needed in such case for an article not to lose its focus. For example, the pathophysiology of and risk factors for atrial fibrillation and venous thromboembolism (does this include deep/superficial venous thrombosis as well as pulmonary embolism?) are quite different from CAD/stroke/aneurysm/PAD. Hemorrhagic stroke and ischemic stroke are also different. Within the current results presentation format (e.g. Table 1), the 15 diseases are listed in a disorganized way, and the discussion are not streamlined to disease entities either (there is a focused paragraph for venous thromboembolism though). Also, transient ischemic attack should be grouped and discussed together with ischemic stroke, not separately.
Response 1: Thank you for the comment. We have reorganized the order of outcomes in Table 1 and diseases have been sorted by three categories: cerebrovascular disease, heart disease and vessel disease. In the Discussion part, we have added several subtitles. A paragraph has been added to supplement the discussion on suggestive associations of FAs with coronary artery disease, aortic valve stenosis and abdominal aneurysm. Considering that atrial fibrillation is a major CVD, a paragraph on atrial fibrillation has been added in the Discussion (Page 8).
“The present study found several suggestive associations between certain FAs and CVDs, such as inverse associations of higher levels of ALA, LA, OA, and POA as well as lower levels of EPA, DPA, DHA, AA, or SA levels with coronary artery disease, aortic valve stenosis and abdominal aneurysm. The observed effects of different FAs on coronary artery disease were overall in line with observational studies [47, 48]. Nonetheless, epidemiological studies focusing on serum FAs and aortic valve stenosis and abdominal aneurysm are limited.
With regard to atrial fibrillation, observational studies consistently concluded a null association between n-3 PUFAs and incidence of atrial fibrillation [49], which is supported by our findings. In addition, among patients undergoing cardiac surgery, supplementation of n-3 PUFAs did not decrease the risk of post-operative atrial fibrillation [50].”
Minor issues:
Point 2: 3rd paragraph (Line numbers after Line 148 are missing) in [Discussion]: you might want to include this updated study as well: Isaksen T, et al. 2019. Thromb Haemost (PMID: 31659738).
Response 2: Thank you for the comment. The references have been updated.
Discussion (Page 8): “and results of a cohort study of Norwegian adults showed that dietary intake of marine n-3 PUFAs was inversely associated with risk of venous thromboembolism [10] and risk of recurrent venous thromboembolism after unprovoked index events [44].”
Point 3: Discussion about atrial fibrillation is totally missing.
Response 3: Most of observational studies did not find any associations between n-3 PUFA and atrial fibrillation. We have added a paragraph on atrial fibrillation in the Discussion part (Page 8).
“With regard to atrial fibrillation, observational studies consistently concluded a null association between n-3 PUFAs and incidence of atrial fibrillation [49], which is supported by our findings. In addition, among patients undergoing cardiac surgery, supplementation of n-3 PUFAs did not decrease the risk of post-operative atrial fibrillation [50].”
Point 4: Line 9 of the last paragraph in Discussion: transferability or generalizability?
Response 4: We have changed to generalizability.
Point 5: Last sentence of conclusion: it is farfetched to reach this conclusion as no therapy has been mentioned throughout the whole article.
Response 5: Thank you for the comment. The last sentence of conclusion has been removed.
Round 2
Reviewer 3 Report
Comments adequately addressed. Much better.